# GenoAgent: A Baseline method for LLM-Based Exploration of Gene Expression Data in Alignment with Bioinformaticians

## Abstract

Recent advancements in machine learning have significantly improved the identification of disease-associated genes from gene expression datasets. However, these processes often require extensive expertise and manual effort, limiting their scalability. Large Language Model (LLM)-based agents have shown promise in automating these tasks due to their increasing problem-solving abilities. To leverage the potential of agentic system, we introduce GenoAgent, a team of LLM-based agents designed with context-aware planning, iterative correction, and domain expert consultation to collaboratively explore gene datasets. GenoAgent provides generalized approach for addressing a wide range of gene identification problems, in a completely automated analysis pipeline that follows the standard of computational genomics. Our experiments with GenoAgent demonstrate the potential of LLM-based approaches in genomics data analysis, while error analysis highlights the challenges and areas for future improvement. We also propose GenoTEX, a benchmark dataset for automatic exploration of gene expression data, and also a promising resource for evaluating and enhancing AI-driven methods for genomics data analysis.

## 1 Introduction

In biomedical research, gene analysis is crucial for understanding biological mechanisms and advancing clinical applications such as disease marker identification and personalized medicine. Advances in next-generation sequencing and other technologies have led to a surge in the volume of transcriptomic data. Genomics research is expected to produce between 2 and 40 exabytes of data in the next decade Institute (2024), greatly facilitating research and discoveries in genomics.

Despite the scientific value of gene data analysis, these tasks are often repetitive, labor-intensive, and prone to errors BPC (2023). The rapid increase in transcriptomic data and potentially inefficient workflows lead to considerable financial burden Intelligence (2023). The genetics research industry incurs an annual expense of around $848.3 million on manual data analysis tasks Research and Markets (2024), with costs expected to increase at a compound annual growth rate (CAGR) of 12% Research and Markets (2024) to 16% Research (2024) by 2030. Bioinformaticians spend significant effort on these repetitive tasks, valued at around $29 per hour Payscale. This high volume of routine tasks greatly impacts job satisfaction among bioinformatics professionals, as surveys show that data scientists, including bioinformaticians, prefer engaging in advanced analytical tasks rather than routine data processing. Currently, up to 45% of their work hours are spent on tasks that could be automated Woodie (2020). These financial and workforce challenges highlight the urgent need for more efficient and cost-effective data analysis solutions in genetics research Bartley (2023).

Meanwhile, the increasing abilities of Large Language Models (LLMs) OpenAI (2024) have enabled methods for automating certain data analysis tasks Ma et al. (2023); Arasteh et al. (2024), and relevant benchmarks have been proposed Stühler et al. (2023); Eldeeb et al. (2024). However, these studies have mostly focused on simplified synthetic datasets, or specific steps in the analyze pipeline such as missing data imputation or hyper-parameter tuning. In contrast, analysis on real-world gene expression data involves complex domain-specific procedures, and inherently requires the

flexible planning, troubleshooting, and domain knowledge inference typically performed by a human bioinformatician, posing higher demands on automatic methods.

To facilitate the development of such methods, we propose **Geno**mics **D**ata **Auto**matic **E**xploration Agents **(GenoAgent)**, a team of LLM-based agents that simulate the behavior of bioinformaticians in gene data analysis. To tackle the challenges in gene data exploration, GenoAgent employs a structured workflow characterized by context-aware planning, iterative correction, and expert consultation, with each agent assigned specific roles that reflect the diverse expertise within a bioinformatics team. By adhering to detailed guidelines, these agents manage the complete data analysis pipeline, from preprocessing to gene identification, thereby streamlining workflows. Our evaluation suggests that GenoAgent is able to automate the process of gene expression data analysis with good overall accuracy, affirming the promise of integrating LLMs into genomics research.

To enhance the evaluation and development of automated gene expression analysis methods, we also propose the benchmark GenoTEX. This dataset facilitates the identification of disease-associated genes while considering biological influences. A trained team of bioinformaticians performed analyses according to these protocols, creating a benchmark dataset comprising input data, annotated code, and analysis outcomes. We define three key tasks—dataset selection, data preprocessing, and statistical analysis—along with metrics to evaluate the automated exploration of gene expression data.

In summary, our contributions are as follows:

- We propose a baseline method, GenoAgent, a team of LLM-based agents to collaboratively explore gene expression datasets. Our evaluation demonstrates the promise of LLM-based approaches in genomics data analysis, and error analysis reveals areas for future improvement.

- We define three challenging tasks: dataset selection, data preprocessing, and statistical analysis, to support more systematic evaluation on performance of GenoAgent.

- We propose a benchmark dataset, GenoTEX, that evaluates the performance of analysis pipeline for a rich set of gene identification problems. We believe it will serve as a useful resource for the evaluation and development of advanced methods for automatic gene expression data analysis.

## 2 RELATED WORK

**LLMs for collaborative problem-solving** Large Language Models (LLMs) have shown the potential to achieve human-level intelligence Wang et al. (2023b); OpenAI (2023); Touvron et al. (2023a;b). Research has tried to enhance their problem-solving abilities through techniques such as goal decomposition Wei et al. (2022); Zheng et al. (2023); Feng et al. (2023); Ning et al. (2023), tree and graph structures Yao et al. (2023); Hao et al. (2023); Besta et al. (2023), consistency Wang et al. (2022b), self-refinement Xi et al. (2023); Madaan et al. (2023); Wang et al. (2023c); Chen et al. (2023), and the use of external tools Liu et al. (2023); Zhao et al. (2023); Qin et al. (2023).

The collaboration of multiple agents can further enhance problem-solving capacities Wang et al. (2023d); Talebirad and Nadiri (2023); Du et al. (2023); Wang et al. (2023a), often through role-playing with distinct expertise Yang et al. (2023a); Dong et al. (2023). MetaGPT Hong et al. (2023) promotes collaboration among various agent roles, and studies have shown the effectiveness of role-playing in software development Qian et al. (2023); Dong et al. (2023). Other works explore sociological phenomena Shapiro et al. (2023); Sumers et al. (2023); Zhou et al. (2023); Wang et al. (2023d); Li et al. (2023), such as virtual towns for interactions among AI agents Park et al. (2023). Recent research emphasizes task management and feedback for performance improvement Huang et al. (2023); Xu et al. (2023); Gou et al. (2023); Yin et al. (2023), with task management shown to enhance multi-agent systems Talebirad and Nadiri (2023); Yang et al. (2023a).

**LLMs for scientific discovery** Researchers have also been incorporating LLMs into scientific discovery in fields such as chemistry Bran et al. (2023); Guo et al. (2023), biotechnology Madani et al. (2023), and medicine Singhal et al. (2023); Yang et al. (2023b) by training or fine-tuning LLMs on domain-specific data. In contrast to these works, we leverage current state-of-the-art LLMs

without additional training. We employ structured prompting and communication strategies to equip LLM-based agents with the planning, analysis, and coding abilities required for scientific exploration.

To tackle the challenging tasks in our benchmark, we propose a baseline method that employs a team of LLM-based agents, each contributing their own expertise, to collaboratively conduct gene expression data analysis.

## 3 METHOD

Recent studies have attempted to leverage LLM-based agents to tackle challenging problems Huang et al. (2023); Yin et al. (2023), including a range of data analysis tasks Ma et al. (2023); Arasteh et al. (2024). While these methods each have their own novelties and strengths, our preliminary experiments reveal that none of them can generate functional code that runs data analysis on gene identification. This is not surprising, considering the full complexity of the analysis required for solving real-world gene data analysis problem, a more tailored approach is probably needed. This section describes our method for exploring and setting up a baseline for this task.

### 3.1 MOTIVATION AND ROLE DESIGN

When human experts engage in complex genomic analysis tasks, they demonstrate several key abilities, including procedural memory, context-aware planning, tool utilization, and domain knowledge inference. We believe that integrating these components is essential for enabling agent systems to navigate the complexities of gene data analysis.

Inspired by the workflows of human bioinformaticians, we propose GenoAgent, a team of LLM-based agents, each equipped with several fundamental features to effectively tackle the challenges of data preprocessing and gene expression analysis.

**Procedural Memory**  Our agent will develop a comprehensive set of guidelines and action sequences for genomic analysis tasks, including optimal parameter selection for data normalization and variant calling. These procedures will be dynamically refined through experience, mirroring the expertise development seen in bioinformaticians. Formally, let $P$ represent the set of procedures and $E$ denote the experience gained. The refinement process can be expressed as $P' = f(P, E)$, where $P'$ is the updated set of procedures and $f$ adjusts them based on accumulated experience.

**Context-aware planning and error corrections**  Before initiating any task, the agent reviews its historical actions and the current genomic analysis context. This review can be formalized as $D(H_t, C_t)$, where $H_t = \{a_1, a_2, \ldots, a_{t-1}\}$ represents the history of actions and $C_t$ represents the current analysis context. This function helps the agent make informed decisions about the next steps, such as adjusting analysis parameters or revising data filters, and to correct any prior errors or inaccuracies. This capability is crucial for ensuring the adaptability and reliability of genomic data analyses.

**Tool Utilization**  Upon deciding on an action, the agent utilizes a curated library of bioinformatics code snippets to perform tasks efficiently. This method is akin to a bioinformatician using well-established bioinformatics libraries. The agent selects the optimal tool by minimizing both time and error, which can be modeled as $T = \text{argmin}_{T_i \in \mathcal{T}} \left( \text{Time}(T_i, \text{task}) + \text{Error}(T_i, \text{task}) \right)$, where $\mathcal{T}$ represents the available tools. If a novel task arises, the agent develops new scripts $T' = \text{GenerateNewTool}(\text{task})$, ensuring both speed and precision in handling complex genomic data.

**Domain Knowledge Inference**  The agent observes the metadata of the dataset and intermediate processing results, using domain knowledge to infer the desired information. This inference process is modeled as $I(K, D) \rightarrow$ True or False, where $K$ is the domain knowledge and $D$ represents the dataset. This allows the agent to check whether their code works as expected, ensuring the accuracy and reliability of their genomic analyses.

The GenoAgent team consists of various specialized roles, each contributing unique expertise to the analysis process. A *Project Manager* coordinates the analysis process for solving each gene identification problem, assigning tasks to agents according to the standardized pipeline from our

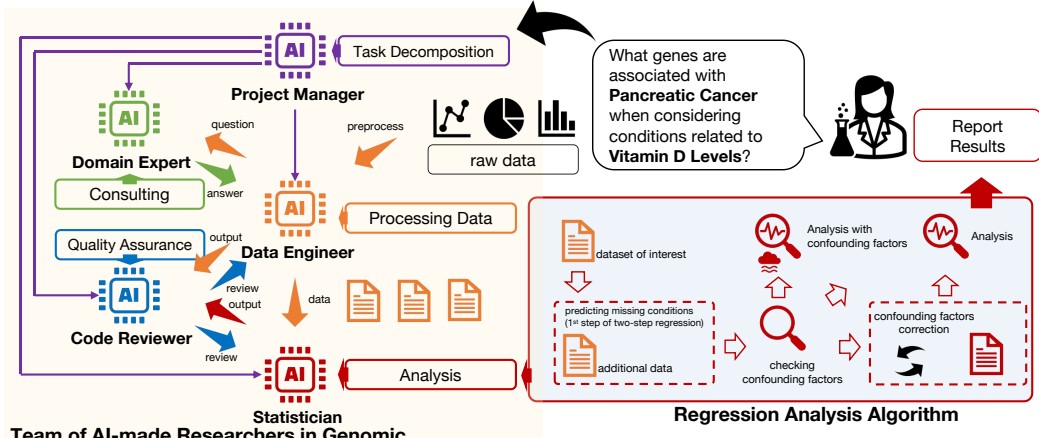

Figure 1: GenoAgent Method Overview

benchmark as instructions; Two programming agents, the *Data Engineer* and the *Statistician*, handle data preprocessing and statistical analysis tasks, respectively. To enable context-aware planning, the agents maintain a task context $C_t = (\text{instruction}, \text{code}, \text{output})$, which records the text instruction, code, and output for each step. Before proceeding, the agents observe the current context $C_t$ and use $D(C_t) \to \{\text{perform}, \text{skip}, \text{revert}\}$ to decide whether to perform the next step, skip it, or revert to a previous one. If writing code is necessary, they can select tools from a function library $\mathcal{L} = \{l_1, l_2, \ldots, l_n\}$. A *Code Reviewer* agents help the programming agents debugging code and verifying that their code follows the instructions. A *Domain Expert* agent provides professional knowledge consultation to programming agents when required for data processing, as shown in Figure 1.

## 3.2 COLLABORATION AMONG LLM AGENTS

This subsection introduces the two main patterns of collaboration between agents.

**Code review and iterative debugging** This process involves the interaction between the Code Reviewer and a programming agent (Statistician or Data Engineer). Let $R(v)$ represent the review function performed by the Code Reviewer, where $v$ is the code version. If the execution of $v$ fails, the reviewer evaluates it based on its execution result, error-free status, and compliance with the instructions. Then the reviewer either approves the code, or rejects it with feedback. Based on the feedback, the programming agent refines the code, representing with $P(v, f)$, where $f$ is the feedback from the reviewer, generating new versions $v_{i+1} = P(v_i, f)$. This process iterates, with the agent generating $v_i$ for $i = 1, 2, \ldots, n$, until either $R(v_n) = $ approved or the maximum debugging rounds $n_{max}$ are reached. This mechanism facilitates troubleshooting and improves adherence to task instructions, as shown in Figure 4 in Appendix.

**Domain-guided programming** The second collaboration pattern involves a Data Engineer consulting a Domain Expert for data preprocessing tasks that require specialized knowledge. The Data Engineer sends questions to the Domain Expert, providing the necessary context such as metadata, summary information about a dataset, or other intermediate results in data processing. Let $D$ represent the dataset and $P$ denote preprocessing functions. The Data Engineer may formulate queries of the form $Q(D)$, seeking $P(D)$. The Domain Expert then provides answers in the form of executable code, as shown in 5 in Appendix. This type of programming also undergoes a debugging process, where execution results $R = P(D)$ are sent back to the same Domain Expert. Some questions are complex enough that the Domain Expert may not provide the correct answer immediately, necessitating further refinement based on the execution results $R$ and adjustments to the preprocessing functions $P$.

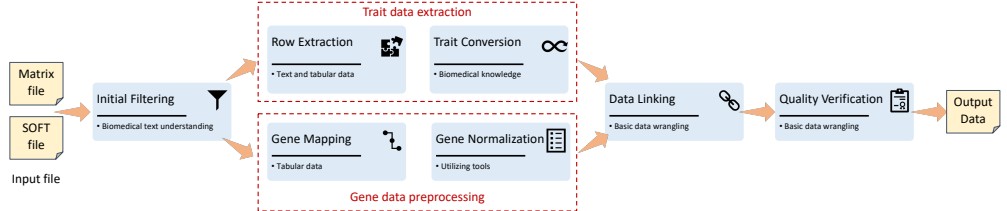

Figure 2: The pipeline for preprocessing a GEO series dataset.

### 3.3 STANDARDIZED PIPELINE FOR GENE EXPRESSION DATA ANALYSIS

Our study aims to automate the gene expression data analysis process to address a class of important problems: *What are the significant genes associated with a specific trait, given the influence of some condition?* Here, a "trait" refers to a characteristic such as a disease (e.g., diabetes), and a "condition" refers to a factor like age, gender, or a co-existing trait (e.g., hypertension). By incorporating these factors into our analysis, we aim to gain a more comprehensive understanding of the genetic underpinnings of these traits.

Thus, to enhance the reliability of our GenoAgent, we have developed a standardized pipeline, serving as an instructive guideline for data preprocessing and statistical analysis tasks, detailed in Appendix A. This pipeline mirrors the steps a skilled bioinformatician would follow, enabling systematic evaluation of the automated methods against established human expertise. In the following subsection, we introduce this pipeline in detail and provide the necessary background knowledge to understand its significance and application in our research.

#### 3.3.1 DATA PREPROCESSING

The preprocessing of gene expression data involves a comprehensive pipeline with several main steps such as dataset filtering and selection, gene data preprocessing, trait data extraction, and data linking. Below we introduce the preprocessing steps for gene expression data within our pipeline. Please refer to our guidelines file in Appendix A for more details. Fig. 2 shows the pipeline of preprocessing a series dataset from the GEO database.

**Dataset filtering and selection** When selecting datasets for gene expression data analysis, the process involves the following steps: (i) **Initial filtering.** We assess each dataset's relevance by reviewing its metadata, ensuring the availability of gene expression data and confirming the traits of interest; (ii) **Quality verification.** Datasets with abnormalities unresolved during preprocessing are discarded to maintain quality; (iii) **Dataset selection.** Given the high dimensionality of gene expression data, we prioritize datasets with the largest sample sizes for single-trait analyses. For two-trait analyses, we select the dataset pair with the highest product of their sample sizes.

**Gene data preprocessing** In this step, we prepare a data table where each attribute represents the expression level of a specific gene within a sample. We map the initial identifiers to gene symbols using platform-specific gene annotation data, then normalize and deduplicate these gene symbols by querying gene databases via APIs to prevent potential inaccuracies due to different gene naming conventions. This process requires flexible planning and proficient use of bioinformatics tools to ensure accuracy and consistency.

**Data linking** In this step, we merge the preprocessed gene data with the extracted trait data based on the sample IDs. This integration creates a data table containing both genetic and clinical features for the same samples, ready for association studies to identify significant genes.

#### 3.3.2 STATISTICAL ANALYSIS

After preprocessing, one can perform basic regression analysis to identify the genes that are predictive of the disease (or trait) Ghosh and Chinnaiyan (2005); Wu et al. (2009). Lasso Tibshirani (1996) is

often chosen as the model due to its ability to identify a sparse set of genes. In addition to directly using regression model, some other steps are often taken.

**Confounding factor correction**    To ensure reliable identification of genes, the pipeline often involves steps to correct potential confounding factors Leek et al. (2010); Bruning et al. (2016). One type of confounding factor arises when the distribution of gene expressions varies across subgroups within the data due to different background distributions rather than the disease itself Yu et al. (2006). This variation can introduce significant bias, leading to incorrect conclusions where the association between certain genes and the disease might be mistakenly attributed to differences in gene expression distributions across groups, rather than a true link to the disease Wang et al. (2022a).

**Incorporating conditions in regression**    Additionally, one can include additional covariates in the regression model, such as patient demographics and co-occurrence of other diseases Kyalwazi et al. (2023). Including these conditions allows for identifying gene expression patterns that are not only associated with the disease status but also modulated by these conditions. This nuanced analysis supports the development of more personalized treatment strategies by identifying how different conditions affect gene-disease relationships Rosenquist et al. (2023). This practice is encouraged due to the need for "precision medicine" Hamburg and Collins (2010); Chan and Ginsburg (2011).

## 4    BENCHMARK

This section describes our GenoTEX benchmark. Specifically, we introduce our process for creating and ensuring the quality of the benchmark, and the tasks and metrics defined for evaluation.

### 4.1    BENCHMARK CREATION

This subsection describes our process of building the benchmark, including the design of gene identification problems, downloading data from open gene expression databases, the collection of manual analysis data, and quality control and assessment.

**Gene identification problem design**    To ensure the scientific relevance of our benchmark, we began by curating a list of human traits that are either important to public health or interesting to genomics research. A computational biologist compiled this list, resulting in 82 traits spanning 9 main categories such as cardiovascular diseases and neurological disorders. This yields 82 problems in the form: *What are the significant genes related to the trait?* (hereafter referred to as "unconditional gene identification").

Next, each trait was paired with a condition, which could be another trait from the list or demographic attributes like age or gender, generating 6806 possible trait-condition pairs. To choose these pairs, we first applied manual criteria based on trait categories (Appendix C). For each undecided pair, we measured trait-condition association by calculating the Jaccard similarity $J(A, B)$ between gene sets $A$

Table 1: Descriptive statistics of our GenoTex benchmark.

| Gene Identification Problems | |
| --- | --- |
| Total problems | 1146 |
| Unconditional problems | 82 |
| Conditional problems | 1064 |
| **Input Dataset** | |
| Total size | 32.22 GB |
| Datasets | 795 |
| Samples per dataset | $167 \pm 121$ |
| Total samples | 132,673 |
| **Manual Analysis and Results** | |
| Relevant datasets | 181 |
| Datasets successfully preprocessed | 163 |
| Lines of code for analyzing per dataset | $90 \pm 32$ |
| Total lines of code for analysis | 71,669 |
| Normalized gene features per dataset | $14174 \pm 5851$ |
| Significant genes identified per problem | $42 \pm 65$ |

(trait) and $B$ (condition) from the NCBI Gene database Brown et al. (2015). Pairs with $J(A, B) > 0.1$ were selected, indicating shared genetic mechanisms valuable for understanding trait-condition interactions. This process identified 1064 pairs of interest, alongside 82 unconditional gene identification problems, forming our benchmark's problem set.

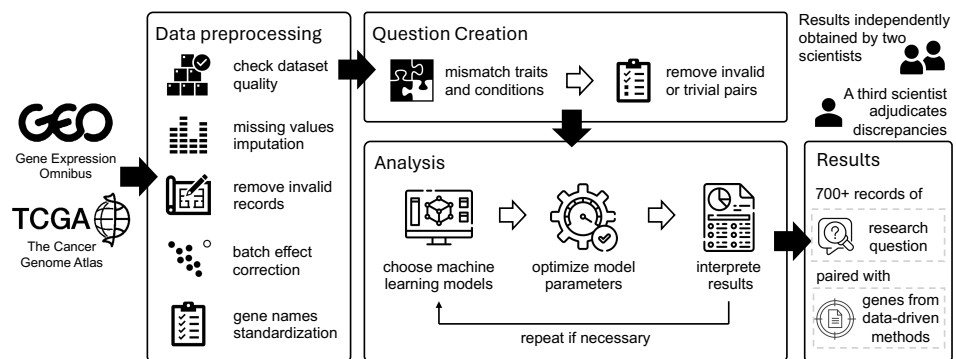

Figure 3: The overview of the GenoTEX benchmark curation.

**Input Dataset**    To address the formulated research problems, we downloaded cohort datasets containing gene expression and corresponding clinical data from public databases: (1) The Gene Expression Omnibus (GEO) Clough and Barrett (2016), the largest gene expression database currently available; and (2) The Cancer Genome Atlas (TCGA) Tomczak et al. (2015), the largest gene expression database focused on cancer. The TCGA data were acquired via the UCSC Xena platform Goldman et al. (2020). Additionally, domain knowledge regarding gene symbols associated with traits was sourced from the NCBI Gene database Brown et al. (2015). For more detailed information about these data sources, please refer to Appendix D.

**Manual analysis**    Four researchers curated the problem list and extracted relevant input data from public sources. In the pilot stage, a computational biologist and a doctoral student developed a guidelines file and example code for solving problems related to two traits, iteratively refining their work based on manual analysis of 200 problems. In the next phase, nine bioinformaticians established a gold standard for analyzing input data across all benchmark problems. This included writing code for data preprocessing and regression analysis. Two researchers analyzed each trait independently, with an experienced researcher adjudicating the annotation by selecting the better analysis and making further refinements, as shown in Figure 3.

To evaluate the consistency of annotations, we measured the Inter-Annotator Agreement (IAA) between the two annotation versions. The results indicate high annotation quality, with an $F_1$ score of 94.73% for the task of dataset filtering. We also used IAA as a baseline for human performance in gene data analysis, with additional results presented in Section 5.

### 4.2 TASKS AND METRICS

**Dataset selection and filtering**    We evaluate the performance of Dataset Filtering and Dataset Selection seperately. The former is a binary classification task, and we use $F_1$ as the primary metric; For the latter, we use accuracy to measure the percentage of problems for which the method chooses the same dataset (or pairs of datasets) as the bioinformations did in our benchmark.

**Preprocessing**    To evaluate the performance of different methods, we adopted the following metrics: (i) Attribute Jaccard (AJ) is the Jaccard similarity between sets of attributes of two datasets. It evaluates how well the method extracts attributes from the dataset by encoding clinical features and normalizing gene symbols. (ii) Sample Jaccard (SJ) is the Jaccard similarity between sets of sample IDs of two datasets. It measures how well the method integrates features of the same samples and handles missing values. Based on these metrics, we define (iii) **Composite Similarity Correlation (CSC)** as the product of the Attribute Jaccard, Sample Jaccard, and the Pearson correlation of the common feature vectors (common rows and columns) between the datasets. This metric captures both the structural and content similarity of the resulting datasets, so we consider it as the primary metric for evaluation preprocessing alignment.

**Statistical analysis**    The goal of statistical analysis is to identify sigificant genes related to traits. To evaluate this process, we adopt multiple metrics such as precision, recall, and Jaccard index. The Jaccard index evaluates the similarity between the sets of genes identified by our method and the

gold standard. We also consider gene identification as a binary classification problem of predicting whether a gene is related to the trait, and use Precision, Recall, and $F_1$ to measure the performance.

## 5 EXPERIMENT

This section describes our experiments to evaluate GenoAgent and other baseline methods on the GenoTEX benchmark. We conducted an end-to-end evaluation where methods process raw input data to complete the full analysis for solving gene identification problems. Additionally, we assessed the performance of each task individually to gain a deeper understanding of their strengths and weaknesses. The tasks and metrics used are defined in Section 4.2. All experiments were conducted on a RunPod cluster RunPod (2024) with two 16-core CPUs and 62 GB RAM. GenoAgent utilizes GPT-4o OpenAI (2024) models accessed via the OpenAI API.

### 5.1 RESULTS

**End-to-end performance**   We evaluated the end-to-end data analysis capabilities of GenoAgent and baseline methods by measuring their performance in gene identification from raw input data. The results in Table 3 show that GenoAgent achieved an $F_1$ score of 51.19%. While this is promising given the task difficulty, there is still a significant gap compared to human inter-annotator agreement scores, indicating substantial room for improvement. Ablation results demonstrated the importance of the collaborative approach involving the Code Reviewer and

Table 2: **Performance of GenoAgent on dataset filtering and selection.** We use $F_1$ and Accuracy for the two subtasks, respectively, where DF stands for Dataset Filtering, and DS stands for Dataset Selection.

| Methods | DF (%) | DS (%) |
|---|---|---|
| GenoAgent (Ours) | **87.32** | **80.25** |
| GenoAgent (Rounds=1) | 85.29 | 76.04 |
| GenoAgent (No Reviewer) | 82.13 | 69.57 |
| GenoAgent (No Domain Expert) | 84.28 | 78.63 |
| Inter-Annotator Agreement | 94.73 | 90.26 |

Domain Expert agents, as well as the number of review rounds. Additionally, we included a simple baseline where GPT-4o was directly asked to answer the significant genes in each problem, resulting in low performance (2.4% $F_1$), which highlights the difficulty of this task. For completeness, we also reported the trait prediction accuracy of the agents' models, reflecting the validity of the data and models they used.

**Dataset filtering and selection**   The performance of dataset filtering and selection is shown in Table 2. The agents show decent performance, likely because determining dataset relevance based on metadata often does not require complex inference. However, errors in this step can propagate to subsequent steps, impacting overall performance.

**Dataset preprocessing**   We evaluated the preprocessing performance of GenoAgent by comparing its output with that of human bioinformaticians in our benchmark. The results are presented in Table 4. GenoAgent generally performed well in preprocessing gene expression and merged data, achieving high CSC scores (80.63% for genes). However, preprocessing of trait data was significantly weaker, with a CSC score of 32.28%, due to the complexity of clinical data extraction and the need for nuanced knowledge inference.

**Statistical analysis**   For the statistical analysis task, we used datasets preprocessed by human bioinformaticians and instructed various baseline methods to perform statistical analysis following our standardized pipeline. The results are shown in Table 5. Unlike data preprocessing, this task primarily involves leveraging Python libraries for generic statistical modeling, allowing several LLMs or agent-based models to achieve decent performance.

### 5.2 DISCUSSIONS

While the results demonstrate the potential of LLM-based methods in gene analysis, they also highlight the limitations of current approaches.

**Instability of the feedback mechanism**   For complex tasks, agents ideally refine their code iteratively based on feedback to reach the correct solution. However, Table 3 shows that while one

Table 3: **End-to-end performance of GenoAgent on the gene identification problems in our benchmark**; additional evaluation on trait prediction performance and the efficiency of LLM API requests for our experiments. Code execution time excluded from the time measurement. We did not include other baseline LLM-as-agent methods such as MetaGPT Hong et al. (2023), because none of them are able to generate runnable code for the preprocessing of gene data, after extensive attempts and given detailed instructions and function tools (Appendix E).

| Methods | Benchmark Performance | | | | Trait Prediction | | | | Efficiency | |
|---|---|---|---|---|---|---|---|---|---|---|
| | Prec.(%) | Rec.(%) | $F_1$(%) | Jac.(%) | Acc.(%) | Prec.(%) | Rec.(%) | $F_1$(%) | Tk.(k) | Time(s) |
| GenoAgent (Ours) | **54.64** | **52.28** | **51.19** | **48.07** | **94.40** | **91.97** | **89.48** | **86.26** | 31.90 | 183.36 |
| GenoAgent (Round=1) | 50.38 | 49.48 | 48.37 | 43.18 | 89.82 | 79.26 | 81.78 | 82.84 | 26.44 | 152.47 |
| GenoAgent (No Reviewer) | 21.35 | 20.20 | 20.10 | 18.77 | 62.81 | 57.76 | 62.58 | 59.31 | 23.85 | 128.63 |
| GenoAgent (No Domain Expert) | 47.94 | 43.80 | 41.33 | 37.19 | 27.82 | 24.68 | 26.59 | 24.79 | 29.23 | 158.37 |
| Inter-Annotator Agreement | 75.58 | 70.64 | 69.66 | 68.64 | - | - | - | - | - | 10.74 |
| GPT-4o zero-shot | 8.47 | 0.12 | 2.41 | 2.69 | - | - | - | - | **0.06** | **8.32** |

Table 4: **Performance of GenoAgent on the preprocessing tasks.**

| Methods | Merged Data | | | Gene Data | | | Trait Data | | |
|---|---|---|---|---|---|---|---|---|---|
| | AJ(%) | SJ(%) | CSC(%) | AJ(%) | SJ(%) | CSC(%) | AJ(%) | SJ(%) | CSC(%) |
| GenoAgent (Ours) | **89.82** | **86.98** | **79.71** | **92.80** | **89.87** | **80.63** | **46.81** | **63.71** | **32.28** |
| GenoAgent (Round=1) | 87.04 | 82.15 | 74.43 | 88.04 | 82.34 | 76.11 | 45.04 | 59.25 | 30.74 |
| GenoAgent (No Reviewer) | 35.18 | 35.06 | 32.73 | 36.01 | 35.7 | 33.62 | 24.02 | 32.58 | 6.45 |
| GenoAgent (No Domain Expert) | 78.54 | 75.93 | 70.01 | 80.79 | 76.38 | 69.67 | 25.14 | 23.48 | 4.68 |

Table 5: **Performance of baseline methods on the statistical analysis task.**

| Methods | Benchmark Performance(%) | | | | Trait Prediction(%) | | | |
|---|---|---|---|---|---|---|---|---|
| | Prec. | Rec. | $F_1$ | Jac. | Acc. | Prec. | Rec. | $F_1$ |
| GenoAgent (Ours) | 68.18 | 62.84 | 67.08 | 68.67 | 57.7 | 57.73 | 58.67 | 57.42 |
| MetaGPT Hong et al. (2023) | 64.90 | 67.20 | 70.28 | 67.14 | 60.63 | 60.85 | 57.04 | 58.55 |
| GPT-4o OpenAI (2024) | 61.61 | 62.75 | 60.48 | 63.85 | 55.39 | 50.72 | 52.50 | 50.42 |
| Llama 3 (8B) Meta (2024) | 8.29 | 10.42 | 8.58 | 12.68 | 8.36 | 8.90 | 5.54 | 5.45 |

feedback round boosts performance compared to none, further rounds yield diminishing returns. Analysis (Appendix F) reveals that Code Reviewer feedback sometimes varies randomly or may be incorrect, contradicting earlier suggestions across multiple rounds, hindering consistent performance. The randomness likely stems from the LLM, highlighting the need to prevent agents from misleading each other. We applied prompt engineering techniques to mitigate this issue(Appendix F), specifically by promoting critical evaluation of feedback in the programming agent and potentially retaining the original code for consistency. Another promising direction is to design collaborative modes where agents iteratively discuss differing opinions to improve task understanding.

# 6 CONCLUSION

In this work, we introduced GenoAgent, a team of LLM-based agents demonstrating the potential of large language models in facilitating the automatic exploration of gene expression data for identifying disease-associated genes. By incorporating mechanism of iterative code review and domain experts programming into standard pipeline, we provide a robust framework for developing and enhancing automated methods. Our experiments highlight both the strengths and limitations of these agents, underscoring the need for further research to address challenges in nuanced human judgment and data anomalies. We also proposed GenoTEX, which is poised to be a useful resource in evaluating and advancing AI-driven genomics data analysis, promoting efficiency, accuracy, and scalability in biomedical research.

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

The supplementary material is organized as follows:

- Appendix A introduces the guidelines file used to standardize the manual curation.
- Appendix B provides examples of manual analysis on trait data extraction.
- Appendix C outlines the criteria for forming trait-condition pairs for gene identification problems in our standardized pipeline.
- Appendix D describes our data acquisition process.
- Appendix E presents our preliminary experiments highlighting the challenges faced by existing LLMs and agent-based methods.
- Appendix F discusses the limitations of GenoAgent.

## A    GUIDELINES FOR GENE EXPRESSION DATA ANALYSIS

To tackle the complexities of gene expression data analysis, we have established a set of comprehensive guidelines shown below. These guidelines try to replicate the detailed processes of a skilled bioinformatician, covering dataset preprocessing, selection, and statistical analysis. By following these standardized procedures, we seek to improve consistency and reliability in our manual benchmark curation.

```
This document describes the standardized pipeline for analyzing gene
expression data for identifying disease-associated genes, involving
dataset preprocessing, selection, and statistical analysis. These
steps follow the practices of computational genomics and ensure the
reproducibility and reliability of the analysis.

Data Sources and Organization:
- Gene expression data are sourced from two public databases,
organized by trait in specific subdirectories:
    - Gene Expression Omnibus (GEO): Data are downloaded under certain
 criteria and saved under the path "{data_root}/GEO". Within this
directory, datasets related to each trait are organized in
subdirectories named after the trait.
    - The Cancer Genome Atlas (TCGA) data via the Xena platform: Data
are saved under the path "{data_root}/TCGA". Similar to GEO, datasets
related to each cancer type are organized in subdirectories named
after the specific cancer trait.

Problem Setting Differentiation:
- If the problem is to identify significant genes predictive of a
trait (optionally conditioning on age or gender, but not involving
another trait), prepare the data related to this trait.
- If the problem is to identify significant genes predictive of a
trait while conditioning on another trait, prepare data for both
traits. These datasets will be integrated in a two-step regression
process.

PART I. GEO Data Preprocessing

Step 1: Initial Data Loading
1. Identify the names of the SOFT file and Matrix file of the Series
data.
2. Read the Matrix file to obtain background information and clinical
trait data. This involves extracting the text data of series titles,
summaries, and overall designs, as well as the tabular data of sample
characteristics.
3. Get the unique values of all attributes in the sample
characteristics table into a Python dictionary.
4. Print the background information and the sample characteristics
dictionary for later observation.
```

**Step** 2: Dataset Analysis and Clinical Feature Extraction
1. Read the metadata to determine if the dataset is likely to contain gene expression data (which does not include miRNA data or methylation data).
2. Based on the metadata and the sample characteristics dictionary, for each of the variables of interest (e.g., a specific trait, age, gender):
    a. Assess the availability of data.
    b. If available, identify the key in the sample characteristics dictionary where unique values of this variable are recorded.
    c. Choose the appropriate data type (continuous, binary, or categorical) and design conversion functions to encode the features into that type.
3. Conduct initial filtering. If either the gene data or trait data is not available, discard this dataset; otherwise, continue with the following steps.

**Step** 3: Gene Data Extraction
1. Read the Matrix file to extract the tabular gene expression data into a dataframe.
2. Print the first few row identifiers in the dataframe for later observation.
3. Determine if the row identifiers are human gene symbols or other types that require mapping.

**Step** 4: Gene Annotation (Conditional)
1. If gene mapping is required, extract the gene annotation table from the SOFT file.
2. Preview the gene annotation table for later observation.

**Step** 5: Gene Identifier Mapping
1. If gene mapping is required, identify the columns for the identifiers and gene symbols from the gene annotation table.
2. Create a mapping dataframe and apply it to the gene expression data. Handle many-to-many relationships between probe IDs and gene symbols by splitting concatenated strings of symbols using separators such as semicolons (;), vertical bars (|), double slashes (//), and commas (,). Assign the corresponding expression values to each gene symbol linked to an identifier. Finally, aggregate the expression values for each gene symbol by averaging the values from multiple probes, with the aim of accurately representing the expression level of each gene symbol.

**Step** 6: Data Normalization and Merging
1. Normalize the gene symbols in the gene data by querying databases with the Python MyGene library, setting the 'scopes' parameter properly. Remove data corresponding to genes that cannot be normalized. For genes that normalize to the same symbol, deduplicate by averaging their expression values.
2. Merge the clinical data with the normalized gene data on sample IDs.

3. Handle missing values. Drop records with the clinical trait missing or with more than 20% of the gene features missing. Use mean imputation for other missing values in the gene expression data.
4. Observe the resulting dataset for quality verification. If the dataset is successfully preprocessed, save the merged data to a CSV file.

PART II. TCGA-Xena Data Preprocessing

**Step** 1: Initial Data Loading
1. Identify the names of the clinical data file and the genetic data file, and load them into two separate dataframes. For gene expression, we choose the 'gene expression RNAseq' dataset instead of its PANCAN normalized or percentile versions.

**Step** 2: Clinical Attribute Selection
1. Print and observe the column names of the clinical data file.
Identify all columns that might hold relevant data for age and gender
from the list of column names.
2. Inspect the first few values of all candidate columns. Select a
single column from the candidate columns that accurately records age
and gender information, respectively, considering meaningful values
and minimal missing data.
3. Based on metadata of the TCGA database, use a simple rule to
convert the trait (whether the sample has the particular type of
cancer) to binary values.
4. Conduct initial filtering. If all samples have the same target
values, or if the clinical dataset shows other abnormalities, discard
the dataset. Otherwise, continue with the next step.

**Step** 3: Data Processing and Merging
1. Normalize the gene symbols in the gene data by querying databases
with the Python MyGene library, setting the 'scopes' parameter
properly. Remove data corresponding to genes that cannot be normalized.
 For genes that normalize to the same symbol, deduplicate by averaging
 their expression values.
2. Merge the clinical and genetic datasets on sample IDs.
3. Handle missing values. Drop records with the clinical trait missing
 or with more than 20% of the gene features missing. Use mean
imputation for other missing values in the gene expression data.
4. Observe the resulting dataset for quality verification. If the
dataset is successfully preprocessed, save the merged data to a CSV
file.

PART III. Statistical Analysis

**Step** 1: Data Selection and Loading
1. Select the best input data relevant to the gene identification
problem, and load the data into a dataframe. If multiple preprocessed
datasets are available for statistical analysis about a trait, we
select the one with the largest sample size.
2. If the analysis requires integrating datasets about two traits, we
sort the possible pairs of datasets for both traits by the product of
their sample sizes, and select the pair with the largest product. Load
 data for the trait and condition into separate dataframes and select
common gene regressors.

**Step** 2: Data Wrangling
1. Extract the relevant data columns and convert into numpy arrays for
 analysis. Get the data matrices of features, the target variable, and
 also the condition when applicable.
2. For two-step regression, this needs to be done twice. In the first
step, the features are the common gene regressors, and the target is
the condition, and we need to extract these matrices from the
condition dataset. The second step follows other cases for extracting
relevant data.

**Step** 3: Condition Prediction (Only for Two-**Step** Regression)
1. Determine the variable type (binary, continuous, or categorical) of
 the condition.
2. Select a simple regression model based on the type of the target
variable, and train it to regress the condition on the common gene
regressors in the condition dataset.
3. Use the trained model to predict the condition values in the trait
dataset using the common gene regressors. Remove the columns in the
trait dataset corresponding to the common regressors, and add the
predicted condition values to it as a new column.

```
Step 4: Model Selection Based on Batch Effect
1. Assess whether the dataset shows batch effects by observing gaps in
 eigenvalues. Choose the appropriate model based on the presence of
batch effects. Use a Linear Mixed Model (LMM) if batch effects are
detected. Otherwise, use a Lasso model.

Step 5: Data Normalization
1. For the feature matrix, and the condition matrix (if applicable),
apply Z-score normalization so that each feature has a mean of 0 and
standard deviation of 1. Make sure this is done every time before
training the model.

Step 6: Hyperparameter Tuning
1. Do 5-fold cross-validation, and perform hyperparameter search on
the logarithm scale with base of 10. Record the best hyperparameter
settings.

Step 7: Model Training
1. Train the model on the entire dataset, with the best
hyperparameters found during cross-validation. For conditional
analyses, incorporate the condition matrix into the model.

Step 8: Model Interpretation
1. Interpret the trained model to identify significant factors and
effects. For Lasso, choose gene variables with non-zero coefficients.
For LMM, apply the Benjamini-Hochberg correction for multiple
hypothesis testing, and select variables whose corrected p-value is
less than 0.05.
2. Save the regression output to a JSON file, with the identified
genes and the corresponding coefficient or p-values.
```

Listing 1: Guidelines file for gene expression data analysis

## B  EXAMPLES OF MANUAL ANALYSIS

In addition to the guidelines file, we provide example files to the participants of our data curation. These examples include code and results for analyzing gene identification problems related to traits such as *Breast Cancer* and *Epilepsy*. These illustrations have proven helpful in familiarizing participants with these tasks quickly. Among the many steps in the analysis pipeline, a key step is the trait data extraction during the preprocessing of GEO data. This step requires biomedical knowledge and an understanding of the dataset collection process described in the metadata. In this section, we will introduce the part of the manual analysis examples related to this crucial step.

### B.1  PROBLEM STATEMENT

Our goal was to extract clinical traits from GEO datasets. For each trait of interest, we aimed to determine its availability and develop encoding rules to automate the extraction process. Below are two examples focusing on *Breast Cancer* and *Epilepsy*, respectively.

### B.2  BREAST CANCER EXAMPLE

#### B.2.1  INPUT DATA

```
!Series_title    "Unlocking Molecular mechanisms and identifying
druggable targets in matched-paired brain metastasis of Breast and
Lung cancers"
!Series_summary  "Introduction: The incidence of brain metastases in
cancer patients is increasing, with lung and breast cancer being the
most common sources. Despite advancements in targeted therapies, the
prognosis remains poor, highlighting the importance to investigate the
 underlying mechanisms in brain metastases. The aim of this study was
```

```
to investigate the differences in the molecular mechanisms involved in
 brain metastasis of breast and lung cancers. In addition, we aimed to
 identify cancer lineage-specific druggable targets in the brain
metastasis. Methods: To that aim, a cohort of 44 FFPE tissue samples,
including 22 breast cancer and 22 lung adenocarcinoma (LUAD) and their
 matched-paired brain metastases were collected. Targeted gene
expression profiles of primary tumors were compared to their matched-
paired brain metastases samples using nCounter PanCancer IO 360 Panel
of NanoString technologies. Pathway analysis was performed using gene
set analysis (GSA) and gene set enrichment analysis (GSEA). The
validation was performed by using Immunohistochemistry (IHC) to
confirm the expression of immune checkpoint inhibitors. Results: Our
results revealed the significant upregulation of cancer-related genes
in primary tumors compared to their matched-paired brain metastases (
adj. p<=0.05). We found that upregulated differentially expressed
genes in breast cancer brain metastasis (BM-BC) and brain metastasis
from lung adenocarcinoma (BM-LUAD) were associated with the metabolic
stress pathway, particularly related to the glycolysis. Additionally,
we found that the upregulated genes in BM-BC and BM-LUAD played roles
in immune response regulation, tumor growth, and proliferation.
Importantly, we identified high expression of the immune checkpoint
VTCN1 in BM-BC, and VISTA, IDO1, NT5E, and HDAC3 in BM-LUAD.
Validation using immunohistochemistry further supported these findings.
 Conclusion: In conclusion, the findings highlight the significance of
 using matched-paired samples to identify cancer lineage-specific
therapies that may improve brain metastasis patients outcomes."
!Series_overall_design    "RNA was extracted from FFPE samples of (
primary LUAD and their matched paired brain metastasis n=22, primary
BC and their matched paired brain metastasis n=22)"
```

Listing 2: Background information for breast cancer

```
{
  0: ['age at diagnosis: 49', 'age at diagnosis: 44', 'age at
diagnosis: 41', 'age at diagnosis: 40', ...],
  1: ['Sex: female', 'Sex: male'],
  2: ['histology: TNBC', 'histology: ER+ PR+ HER2-', 'histology:
Unknown', 'histology: ER- PR- HER2+', 'histology: ER+ PR-HER2+', '
histology: ER+ PR- HER2-', 'histology: ER- PR+ HER2-', 'histology:
adenocarcinoma'],
  3: ['smoking status: n.a', 'smoking status: former-smoker', 'smoking
 status: smoker', 'smoking status: Never smoking', 'smoking status:
unknown', 'smoking status: former-roker'],
  4: ['treatment after surgery of bm: surgery + chemotherpy', '
treatment after surgery of bm: surgery + chemotherpy + Radiotherapy', '
treatment after surgery of bm: surgery + chemotherapy + Radiotherapy',
 'treatment after surgery of bm: surgery', 'treatment after surgery of
 bm: surgery + chemotherapy + Radiotherapy', ...]
}
```

Listing 3: Sample characteristics for breast cancer. Some long lists are truncated for brevity.

### B.2.2 INFERENCE PROCESS

The dataset summary indicated that tissue samples from primary breast cancer (BC) and lung adeno-carcinoma (LUAD), along with their matched-paired brain metastases, were included. By examining the sample characteristics dictionary, combined with domain knowledge, we identified subtypes such as 'TNBC', 'ER+', 'PR+', and 'HER2+' associated with breast cancer, and 'adenocarcinoma' associated with lung cancer. Based on this, we developed a rule: tissues labeled with 'TNBC', 'ER+', 'PR+', or 'HER2+' are coded as having breast cancer (1), while 'adenocarcinoma' is coded as not having breast cancer (0).

```
def convert_trait(value):
```

```
     if 'TNBC' in value or 'ER+' in value or 'PR+' in value or 'HER2+'
in value:
         return 1  # Breast Cancer
     elif 'adenocarcinoma' in value:
         return 0  # Not Breast Cancer (LUAD)
     else:
         return None  # Unknown
```

Listing 4: Python function to encode Breast Cancer trait

### B.3 EPILEPSY EXAMPLE

#### B.3.1 INPUT DATA

```
!Series_title    "Integrated analysis of expression profile and
potential pathogenic mechanism of temporal lobe epilepsy with
hippocampal sclerosis"
!Series_summary  "To investigate the potential pathogenic mechanism of
 temporal lobe epilepsy with hippocampal sclerosis (TLE+HS), we have
employed analyzing of the expression profiles of microRNA/ mRNA/
lncRNA/ DNA methylation in brain tissues of hippocampal sclerosis (TLE
+HS) patients. Brain tissues of six patients with TLE+HS and nine of
normal temporal or parietal cortices (NTP) of patients undergoing
internal decompression for traumatic brain injury (TBI) were collected.
 The total RNA was dephosphorylated, labeled, and hybridized to the
Agilent Human miRNA Microarray, Release 19.0, 8x60K. The cDNA was
labeled and hybridized to the Agilent LncRNA+mRNA Human Gene
Expression Microarray V3.0, 4x180K. For methylation detection, the DNA
 was labeled and hybridized to the Illumina 450K Infinium Methylation
BeadChip. The raw data was extracted from hybridized images using
Agilent Feature Extraction, and quantile normalization was performed
using the Agilent GeneSpring. We found that the disorder of FGFR3, hsa-
miR-486-5p, and lnc-KCNH5-1 plays a key vital role in developing TLE+
HS."
!Series_overall_design    "Brain tissues of six patients with TLE+HS
and nine of normal temporal or parietal cortices (NTP) of patients
undergoing internal decompression for traumatic brain injury (TBI)
were collected."
```

Listing 5: Background information for Epilepsy

```
{
  0: ['tissue: Hippocampus', 'tissue: Temporal lobe', 'tissue:
Parietal lobe'],
  1: ['gender: Female', 'gender: Male'],
  2: ['age: 23y', 'age: 29y', 'age: 37y', 'age: 26y', 'age: 16y', 'age:
 13y', 'age: 62y', 'age: 58y', 'age: 63y', 'age: 68y', 'age: 77y', '
age: 59y', 'age: 50y', 'age: 39y']
}
```

Listing 6: Sample characteristics for Epilepsy

#### B.3.2 INFERENCE PROCESS

The dataset summary indicated that brain tissues from patients with temporal lobe epilepsy with hippocampal sclerosis (TLE+HS) and control samples were included. By examining the sample characteristics dictionary, we identified tissue types such as 'Hippocampus', 'Temporal lobe', and 'Parietal lobe'. We inferred that 'Hippocampus' and 'Temporal lobe' tissues are associated with TLE+HS (epilepsy), while 'Parietal lobe' tissues are from control samples. Based on this, we developed a rule: tissues labeled with 'Hippocampus' or 'Temporal lobe' are coded as having epilepsy (1), while 'Parietal lobe' is coded as control (0).

```python
def convert_trait(value):
    if 'Hippocampus' in value or 'Temporal lobe' in value:
        return 1  # Epilepsy (TLE+HS)
    elif 'Parietal lobe' in value:
        return 0  # Control (NTP)
    else:
        return None  # Unknown
```

Listing 7: Python function to encode Epilepsy trait

### B.4 VALIDATION AND CONCLUSION

By executing the provided Python functions, we confirmed the accuracy of our trait extraction process. For instance, applying the `convert_trait` function for the epilepsy dataset, we verified the presence of exactly six samples with the positive *Epilepsy* trait, consistent with the metadata description. Similarly, for the breast cancer dataset, the function accurately identified 22 samples with the *Breast Cancer* trait. These examples highlight the dataset context understanding and domain knowledge inference required for the accurate preprocessing of gene expression data.

## C CRITERIA FOR MANUAL CORRECTION OF TRAIT-CONDITION PAIRS

To ensure the scientific validity of our benchmark questions, we apply specific rules for including and excluding certain trait-condition pairs. Each biomedical entity in our list can be considered a trait and paired with a condition, where the condition is either another entity from the list or a demographic attribute like "age" or "gender." The following criteria are designed to maintain scientific relevance and robustness:

- **Trait-Condition Role Assignment**: Entities such as language abilities, Vitamin D levels, and bone density are included only as conditions and not as traits. This distinction ensures that the primary focus remains on traits with more direct clinical implications, while these entities serve as influential factors that could affect those traits.
- **Universal Conditions**: Entities such as obesity, hypertension, and mental disorders like anxiety disorder and bipolar disorder are designated as conditions to be paired with all other traits. This is because these conditions are widespread and significantly impact various health outcomes, making them critical factors to consider in any genetic analysis.
- **Gender-Specific Considerations**: Gender-specific entities such as prostate cancer, endometriosis, and breast cancer are not conditioned on gender. Furthermore, entities from different genders are not paired. This approach respects the biological distinctions between genders and ensures that the resulting questions remain relevant and meaningful.
- **Cancer Category Exclusion**: Pairs where both the trait and the condition belong to the cancer category are excluded. This is because investigating genetic factors behind one type of cancer conditioned on another type of cancer is often less scientifically important. The focus is placed on broader, more impactful genetic relationships that offer greater insight into cancer biology.

These criteria are used in combination with the Jaccard similarity of related genes (Section 3.2), to uphold the scientific integrity and relevance of the benchmark questions, facilitating meaningful and insightful gene expression analysis.

## D DETAILS ABOUT THE DATA SOURCES

**GEO** The Gene Expression Omnibus (GEO) (Clough and Barrett, 2016) is a public archive for high-throughput gene expression data and various other types of genomic data. We leveraged the Entrez programming utility to perform a systematic search of the GEO database for human series data relevant to each trait on our list, prioritizing datasets with large sample sizes. We downloaded both SOFT and matrix files for each series and used heuristic evaluations of file sizes to pinpoint

datasets likely containing gene expression data. When automated searches failed to yield results for specific traits, we conducted manual searches using expanded synonyms from Medical Subject Headings (MeSH) terms.

**TCGA-Xena**   The Cancer Genome Atlas (TCGA) (Tomczak et al., 2015), accessed through the Xena platform Goldman et al. (2020), offers a rich repository of RNAseq gene expression and clinical data for numerous cancer types. We obtained data for 36 traits from the TCGA cohort using the UCSC Xena platform, which provides high-quality, cancer-related gene expression and clinical data linked by patient IDs.

**NCBI Gene**   The NCBI Gene database (Brown et al., 2015) is an important resource for comprehensive information on gene sequences, functions, and their links to diseases and conditions. For each trait, we queried the database to compile sets of gene symbols associated with the trait. This data was crucial for identifying disease-disease associations for question generation and for selecting common regressors in two-step regression analyses.

# E   CHALLENGES FACED BY EXISTING METHODS ON THE GENOTEX BENCHMARK

Gene expression data analysis is a complex and specialized task. Despite their problem-solving abilities, state-of-the-art LLMs and agent-based methods struggle with gene expression data. Our evaluations of methods such as GPT-4o OpenAI (2024), MetaGPT Hong et al. (2023), and CodeAct Wang et al. (2024) revealed consistent failures across various settings.

We tested these methods under three different settings: (i) providing general task instructions, (ii) providing detailed task instructions used by GenoAgent, and (iii) providing detailed task instructions and all necessary library functions as in GenoAgent. Each setting was tested on a subset of 50 gene identification problems. Our results show that none of the methods generated runnable code for preprocessing datasets downloaded from GEO. Persistent errors in the generated code prevented testable outputs, regardless of the level of detail provided.

First, we find that when preprocessing GEO data, these methods often fail at dataset loading in the initial steps. The gene expression data files follow special formats. The agent struggles to extract tabular data embedded in the text file by identifying special markers, skipping metadata rows, and setting other parameters correctly, resulting in data reading failures.

```python
import pandas as pd
from typing import Tuple
from utils import Utils

class DataLoader:
    """
    DataLoader class is responsible for loading clinical and genetic
    data from given file paths.
    """

    def load_clinical_data(self, filepath: str) -> pd.DataFrame:
        """
        Loads clinical data from a specified file path.

        :param filepath: The path to the clinical data file.
        :return: A pandas DataFrame containing the clinical data.
        """
        try:
            clinical_data = pd.read_csv(filepath)
            Utils.log(f"Clinical data loaded successfully from {filepath}")
            return clinical_data
        except FileNotFoundError:
            Utils.log(f"File not found: {filepath}")
            raise
```

```
        except pd.errors.EmptyDataError:
            Utils.log(f"No data: {filepath} is empty")
            raise
        except Exception as e:
            Utils.log(f"An error occurred while loading clinical data:
{e}")
            raise
```

Listing 8: Failure example of MetaGPT in reading datasets

We manually corrected the data loading code for the baseline methods and continued with the tasks. However, they were still unable to conduct the inference required to extract clinical features. This step is inherently difficult and often requires at least one round of debugging by the Domain Expert agent in our GenoAgent method to achieve a higher success rate.

```
def convert_trait(self, value: str) -> str:
    """
    Converts a trait value to a standardized string format.

    :param value: The trait value to convert.
    :return: A standardized string representation of the trait.
    """
    # This is a placeholder for the actual conversion logic, which would
    # depend on the specific requirements for trait conversion.
    # For example, it could map various synonyms to a canonical form.
    standardized_value = value.strip().lower()
    return standardized_value
```

Listing 9: Failure example of MetaGPT in encoding Breast Cancer trait

```
def convert_trait(value):
    if value in ['TLE+HS', 'control']:
        return 1 if value == 'TLE+HS' else 0
    return None
```

Listing 10: Failure example of CodeAct in encoding Breast Cancer trait. 'TLE+HS' is indeed related to epilepsy according to the metadata, but this is not the way the trait information is recorded for each sample. Moreover, these functions didn't strip the content before the colon. As a result, the code will convert all trait values to None.

The challenges faced by methods like MetaGPT and CodeAct in processing gene expression data primarily stem from their difficulty in handling specialized data formats and the absence of flexible feedback mechanisms. MetaGPT, primarily designed for software engineering tasks, operates with an independent execution model and limited context-awareness, which can impede dynamic adaptation during task execution and lead to errors when dealing with the nuanced formats of gene expression datasets. CodeAct, while effective at generating executable code through structured prompts, lacks the context-aware planning and iterative refinement necessary for the intricate steps involved in gene expression data preprocessing. Its static approach does not easily accommodate the dynamic adjustments required for diverse and complex gene expression data, leading to errors during initial data loading and clinical feature extraction.

In contrast, GenoAgent employs a team of specialized agents that maintain a comprehensive task context and leverage expert consultation, allowing for context-aware planning and iterative correction. This enables GenoAgent to handle the complexities of genomics data analysis more effectively, improving its reliability in data preprocessing.

## F    DISCUSSION ON THE LIMITATIONS OF GENOAGENT

This section discusses the observed limitations of our baseline method, GenoAgent, on the GenoTEX benchmark. We identified that certain steps are inherently challenging, and instability in the feedback

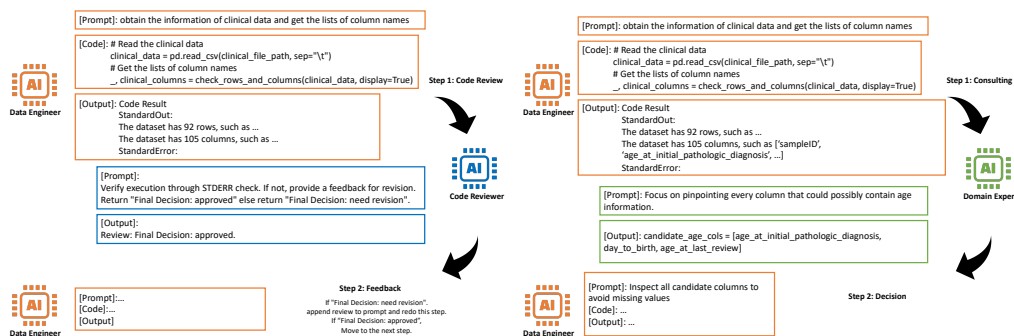

Figure 4: The collaboration between Data Engineer and Code Reviewer.

Figure 5: The collaboration between Data Engineer and Domain Expert.

mechanism may hinder the agents' iterative improvement process. Figures 4 and 5 illustrate the two types of feedback mechanisms in GenoAgent.

## F.1 ERROR EXAMPLE IN PREPROCESSING

The results in Table 4 of the main paper indicate that the preprocessing performance of GenoAgent is primarily constrained by the clinical feature extraction step, which shows a CSC of only 32.28%. This step is conducted through Domain-Guided Programming (Section 4.2), where the Domain Expert iteratively improves its output based on feedback from the execution environment. Although one round of self-review significantly enhances performance, increasing the maximum review rounds from 1 to 2 yields only marginal benefits. Detailed examination of the agent system's operation log at this step across different experimental runs reveals that the Domain Expert's answers to the same question can vary randomly.

For example, consider the following function used to encode the Breast Cancer trait:

```python
def convert_trait(value):
    if 'breast cancer' in value.lower():
        return 1
    elif 'lung adenocarcinoma' in value.lower():
        return 0
    else:
        return None
```

Listing 11: Failure example of GenoAgent for encoding Breast Cancer trait

In one run, the code review provided the following feedback:

```
*Issue*: The convert_trait function assumes that the input string will
 always mention either "breast cancer" or "lung adenocarcinoma," which
 might not always be the case. The current data may not explicitly
have such a field.

*Suggestion*: Update the function to be more flexible by incorporating
 biomedical knowledge. Since the dataset deals explicitly with breast
cancer and lung adenocarcinoma, we can assume breast cancer is present
 based on the context or use a default binary value.

*Final Decision*: Approved
```

Listing 12: Code review that didn't correctly find the issue, and approved the code

However, in another run with the identical setting, the code review provided different feedback:

```
*Issue*: The convert_trait function does not conform to the
instructions. The traits should be inferred from the histology field.
```

```
*Suggestion*: Adjust the function to check for breast cancer subtypes
in the histology field. The current implementation checks for "breast
cancer" and "lung adenocarcinoma" in the value, which might not
directly map to the provided sample characteristics. A more context-
aware approach, utilizing the provided histology information to infer
the presence of breast cancer, would be more effective.

*Final Decision*: Rejected
```

Listing 13: Another run of code review, which correctly analyzed the issue and rejected the code

### F.2 Error example in statistical analysis

Analysis of failure cases during the statistical analysis task reveals various low-frequency random failures, including errors in extracting data matrices from dataframes and incorrect parameter passing to the regression model. Although no single bottleneck was identified, the cumulative error risk significantly impacts performance, resulting in a suboptimal $F_1$ score of 67.08%. This task involves collaboration between the Statistician and Code Reviewer (Section 4.2). Similar to preprocessing, we observed unstable and inconsistent feedback from the Code Reviewer.

The following is an example of erroneous code generated by the Statistician agent:

```python
from utils.statistics import *

# Set directory and trait, omitted here.

# Step 1: Load input data about the trait into a dataframe
trait_data, _, _ = select_and_load_cohort(data_root, trait, condition=
None, is_two_step=False)

# Step 2: Extract X and Y for regression analysis
Y = trait_data[trait].values
X = trait_data.drop(columns=[trait]).values

# Step 3: Selecting appropriate model based on batch effect
batch_effect_detected = detect_batch_effect(X)
model_constructor = LMM if batch_effect_detected else Lasso

# Step 4: Hyperparameter search
param_values = [10 ** i for i in range(-6, 1)]  # powers of 10 from 1e
-6 to 1e0
var_names = list(trait_data.columns)
if batch_effect_detected:
    fixed_params = {"fit_intercept": False}
else:
    fixed_params = {"fit_intercept": True, "max_iter": 10000}
best_config, _ = tune_hyperparameters(
    model_constructor, param_values=param_values, X=X, Y=Y, var_names=
var_names,
    trait=trait, gene_info_path=gene_info_path, condition=None, Z=None,
 fixed_params=fixed_params
)

# Step 5: Normalize X
X_normalized, _ = normalize_data(X)

# Step 6: Train the model with best hyperparameter on whole dataset
model = ResidualizationRegressor(model_constructor, best_config)
model.fit(X_normalized, Y)

# Step 7: Interpret trained model for significant genes and save
results
```

```
significant_genes = interpret_result(model, var_names, trait,
condition=None, print_output=True)

os.makedirs(os.path.join(output_dir, normalize_trait(trait)), exist_ok
=True)
save_path = os.path.join(output_dir, normalize_trait(trait))
save_result(
    significant_genes, output_dir=save_path, condition=None
)

[Output]:

[Execution Error]:
__init__() got an unexpected keyword argument 'fit_intercept'
```

Listing 14: Failure example of the Statistician agent

**Discussion**   The randomness observed may stem from the LLM itself, suggesting a need to prevent one agent from misleading another. During the development of our baseline methods, we implemented several prompt engineering techniques to mitigate this issue: (i) Limiting the Reviewer's feedback to three main suggestions to focus on problem-solving rather than providing numerous distracting comments about code quality, and (ii) Encouraging the agent receiving the review to critically evaluate the feedback and possibly retain its original code. While these measures have alleviated some issues, they persist to some extent in our GenoAgent baseline. A promising future direction involves designing collaborative modes that foster iterative discussions among agents to reconcile differing opinions and enhance their task performance abilities.

We hope this discussion highlights the challenges of our benchmark tasks and encourages future work to address these issues.

