# OpenReview forum: "GenoAgent: A Baseline method for LLM-Based Exploration of Gene Expression Data in Alignment with Bioinformaticians"
_ICLR.cc/2025/Conference — ICLR 2025 Conference Withdrawn Submission_

### Official Review · Reviewer_7uNF · 2024-10-30

**Soundness:** 2
**Presentation:** 2
**Contribution:** 2
**Rating:** 3
**Confidence:** 5

**Summary:**

The authors in this paper present GenoAgent which comprises of LLM-based agents that are responsible for different roles. They also present a benchmark, GenoTEX for automatic exploration of gene expression data and use this to assess different tasks such as automation in genomics. They report F1, Precision, Recall, Accuracy and Jaccard similarity between gene sets (trait and condition).

**Strengths:**

S1: The authors use different agents that are responsible for different roles in the GenoAgent pipeline. These agents are LLM-based and the authors have clearly defined the responsibilities each “agent” would undertake such as developing guidelines or reviewing historical actions and current context.
S2: There are specialized roles in the pipeline as well – “Project Manager”, “Data Engineer”, “Statistician”, “Code Reviewer” and “Domain Expert”. These roles bring about modularity in GenoAgent which in turn would help with issues like troubleshooting.
S3: The creation of the GenoTex benchmark is novel in its contribution to the genomic data analysis since it comprises of gene identification problems, data from open gene expression databases, manual analysis data, quality control and assessment.

**Weaknesses:**

W1: The performance metrics, especially using F1 and accuracy are not very expressive in understanding just how good the pipeline is. F1 can sometimes not fully capture data filtering requirements and this is very crucial in genomics.
W2: There are several ambiguous statements, ex: “the process of gene expression data analysis with good overall Accuracy” or “The agents show decent performance, likely” which do not correctly reflect the accuracy or performance.
W3: The feedback mechanism does not seem entirely reliable in the Code Reviewer agent, especially when it shows diminishing performance after the first feedback round. This issue is quite critical when it comes down to understanding the robustness of GenoAgent.
W4: Data normalization can be further refined to reduce variability for traits.
W5: While GenoTEX benchmark is novel, the authors do not provide an exhaustive comparison with any existing domain-specific methods.
W6: What are the computational costs for using computationally expensive models like GPT-4o and other LLMs? To add to this, all the agents are LLM-based. Have the authors kept this in check?

Typos:
1. Line 57, why are “D” in “Data”, “Auto” in “Automatic” and “E” in “Exploration” made bold when it is not used in the abbreviation “GenoAgent”?

References:
1. Line 501 (part of the url) is spilling outside the margin.
2. Incorrect citations for several references (ex: lines 496, 521, 540, 575, 579, 591, 658, 694 ). I have listed some of the corrected references at the bottom (Harvard format) to give the authors an example.
3. Citation format inconsistent (and possibly incorrect) in line 593.
4. There is inconsistency in the format for the references in general.
5. Medium articles are often unchecked, unverified facts wherein the scientific rigor can be easily questioned. Authors have cited some, like “BPC (2023)” on line 38/39.

Line 496: Besta, M., Blach, N., Kubicek, A., Gerstenberger, R., Podstawski, M., Gianinazzi, L., Gajda, J., Lehmann, T., Niewiadomski, H., Nyczyk, P. and Hoefler, T., 2024, March. Graph of thoughts: Solving elaborate problems with large language models. In Proceedings of the AAAI Conference on Artificial Intelligence (Vol. 38, No. 16, pp. 17682-17690).
Line 521: Dong, Y., Jiang, X., Jin, Z. and Li, G., 2024. Self-collaboration code generation via chatgpt. ACM Transactions on Software Engineering and Methodology, 33(7), pp.1-38.
Line 540: Guo, T., Nan, B., Liang, Z., Guo, Z., Chawla, N., Wiest, O. and Zhang, X., 2023. What can large language models do in chemistry? a comprehensive benchmark on eight tasks. Advances in Neural Information Processing Systems, 36, pp.59662-59688.
Line 575: Ma, P., Ding, R., Wang, S., Han, S. and Zhang, D., 2023, December. InsightPilot: An LLM-empowered automated data exploration system. In Proceedings of the 2023 Conference on Empirical Methods in Natural Language Processing: System Demonstrations (pp. 346-352).

**Questions:**

Q1: How reliable is prompt engineering since it does not really eliminate the possibility of hallucinating?
Q2: Can improving the preprocessing of clinical data improve the performance? If yes, what strategies have the authors considered?

---

### Official Review · Reviewer_Atae · 2024-11-03

**Soundness:** 3
**Presentation:** 2
**Contribution:** 2
**Rating:** 5
**Confidence:** 3

**Summary:**

The paper presents GenoAgent, a novel framework leveraging a team of LLM-based agents to automate the exploration and analysis of gene expression data, addressing challenges in scalability and expertise demands in genomics. Each agent in GenoAgent has a specialized role—such as data engineering, statistical analysis, code review, and domain expertise—mimicking a collaborative bioinformatics workflow for tasks like dataset selection, preprocessing, and statistical analysis. Additionally, the authors introduce GenoTEX, a benchmark dataset designed for evaluating and advancing automated methods in gene analysis. Experiments demonstrate that GenoAgent achieves promising accuracy in gene identification, with iterative error correction and expert consultation mechanisms enhancing its overall performance, while GenoTEX provides a resource for consistent, real-world evaluation of automated genomics tools.

**Strengths:**

1.	This paper is among the first to apply an LLM-based multi-agent system to gene expression data analysis, simulating collaborative workflows typical in bioinformatics teams.

2.	Dataset Contribution: The GenoTEX benchmark provides a valuable resource for future research in AI-driven genomics data analysis, offering a standardized framework for evaluating model performance.

**Weaknesses:**

1. GenoAgent has been tested on benchmark datasets, but applying it to novel, unseen datasets in real-world genomics research to assess its robustness under different conditions would enhance the effectiveness of this work.

2. While GenoAgent introduces a structured, team-based approach to LLM-driven gene expression analysis, the novelty of its methodology could benefit from deeper differentiation from existing frameworks in multi-agent LLM systems and automated bioinformatics workflows.

**Questions:**

1. The paper mentions that feedback from the Code Reviewer is sometimes inconsistent. How frequent are these inconsistencies, and how do they typically impact GenoAgent’s task outcomes? Could the authors provide data on the frequency of conflicting or erroneous feedback in the code review process that leads to downstream errors in the analysis?

2. A comparative table or discussion on how GenoAgent specifically improves upon or extends prior work would help emphasize its novelty.

3. A discussion of runtime, memory usage, or potential computational optimizations (such as modularizing tasks or limiting interactions between agents) would aid in understanding the feasibility of GenoAgent for widespread use. Since the analysis pipeline can be resource-intensive, identifying strategies to minimize costs would be crucial for scaling GenoAgent in practical environments.

4. While the paper includes error analysis, it could benefit from a deeper examination of errors in statistical analysis and preprocessing steps. Are there specific types of errors (e.g., issues with gene symbol normalization or confounding factor adjustments) that are more prevalent, and what are the proposed fixes?

---

### Official Review · Reviewer_jvkF · 2024-11-03

**Soundness:** 2
**Presentation:** 2
**Contribution:** 2
**Rating:** 3
**Confidence:** 5

**Summary:**

They introduce GenoAgent, a team of LLM-based agents designed with context-aware planning, iterative correction, and domain expert consultation to explore gene datasets.

**Strengths:**

1. Proposed an agent system to explore gene expression datasets.
2. Proposed a benchmark dataset, GenoTEX.

**Weaknesses:**

The experiments are not enough.

**Questions:**

1. Have you tried your system with different backbones? You have claimed that you use GPT-4o as the backbone LLM. I’d like to know the performance of Llama3-8B or other open-source models.
2. How could you ensure no data leakage on the gene identification problems? Could you show the comparison experiments between your agent and other LLMs? You have claimed that MetaGPT cannot generate runnable code for the preprocessing of gene data. How could you make sure that your model could generate runnable code? Only by iterations and reviews? In addition, since the code reviewer is still based on LLMs, this agent is also based on the ability of LLMs. It doesn’t contain any fine-tuning stages, so how could you make sure your agent could generate runnable code?
3. How about the success rate for generating code?
4. In addition, the authors have claimed a set of function library L. Is this process automatic? Or just by prompting LLMs?

---

### Official Review · Reviewer_KiDk · 2024-11-04

**Soundness:** 4
**Presentation:** 4
**Contribution:** 3
**Rating:** 5
**Confidence:** 3

**Summary:**

This work introduces GenoAgent and LLM agent-based framework for gene expression
analysis tasks. This framework mainly consists of a project manager agent,
domain expert, data engineer, code reviewer and statistician agents.
Generally, the method is able to solve trait-related questions from raw data
by leveraging statistical (code-based) tools. To evaluate the proposed
method, the authors curate the GenoTEX benchmark, a benchmark of unconditional
and conditional pairs of traits genes, which is also manually adjusted by experts.
The results generally show great performance of the proposed method in the
tasks.

**Strengths:**

The paper is generally well-written, and in good English. The motivation
of the problem is clearly stated and the related work is well covered.
The method proposed is really innovative and promising. Additionally,
the authors define carefully the tasks undergone to curate the dataset
and the steps reported seem reasonable. Generally, the results are positive
and I believe that the paper has the potential to be a great contribution
to the field.

**Weaknesses:**

I am mainly concerned about two major issues in the paper:

- The first one is regarding the availability and clarity of the reported
methods. The authors claim that the GenoTEX benchmark will be a great
contribution to the field. Does this mean that the benchmark will be
publicly available? If so, it would have been great for the authors
to provide (if possible) an anonymized link to the benchmark.
Regarding source code, the authors do not mention if the source code
will be available and no link/supplementary material is provided.
Again, it would be very positive for the authors to release
the source code for the proposed method, as this would allow for
the reproducibility of the results. Otherwise, there are parts of the
work that might seem obscure to the reader, e.g. what are the statistical
tools used by the agents and how? How are the agents and their communication implemented?, etc.

- The second issue is regarding the clarity of the results. The authors
provide a good description of the curation process of the benchmark,
however, the results provided are not very detailed, which makes it
hard for the reader to understand wether the authors achieved the
goals of the paper and how well the proposed method performs.
I would suggest the authors to provide a more clear, detailed and
a comprehensive description of the results, including a more detailed
descriptions of the tasks at hand, the significance of the metrics used
and to provide insightful comments that could serve the community
or the reader of the paper. Please see the questions below for a few examples
of the issues regarding the clarity of the results.

**Questions:**

This is a list of minor comments, questions and suggestions:

- On line 351: "an experienced researcher adjudicating the annotation by selecting the better analysis and making further refinements".
What do these "annotations" refer to? I assume that these doesn't refer to gene annotations
(correct me if wrong). Without this information is really hard to evaluate what "Inter-Annotator Agreement (IAA)" refers to.

- On line 418, "complexity of clinical data extraction and the need for nuanced knowledge inference". Could you please elaborate more on what does this mean?

- On line 424, " allowing several LLMs or agent-based models to achieve decent performance".
It seems to me that in the previous tasks, different models could also perform differently. Please, correct me if wrong.

- What does "merged data" refer to in Table 4?

- In Table 5, what is really the difference between GenoAgent (based on GPT-4o) and GPT-4o itself?
I understand that the former doesn't have agents involved, is this correct?

- It would have been great to comment on the "Statistical analysis" results reported.
What were the expected results? Are the results reported positive?

- Is "Dataset selection and filtering" with GenoAgent performed entirely with metadata
from the datasets only?

- On line 395: "measuring their performance in gene identification from raw input data"
What does "raw input data" refer to here?

- I am assuming that the goal of "end-to-end performance" in Section 5.1
is to measure the performance in identifying significant genes related to traits.
Then I assume that this is a multi-label classification problem. Is this correct?
If so, how many classes are considered for the results? How imbalanced is the dataset?

- From Table 5, is there any advantage here to using GenoAgent against the other models?

- On line 417: "However, preprocessing of trait data was significantly
weaker, with a CSC score of 32.28%, due to the complexity of clinical data extraction and the need for nuanced knowledge inference"
Could you please elaborate more on this?

- What is the total number of datasets in the results of Table 2?

- Table 2 says that it reports "F1 and Accuracy" for "DF" and "DS", which one of the two is reported?

- About the results reported in Table 4, I assume that the results are indeed good,
but without any context or reference it is hard to evaluate whether CSC of 79.71 is
a good or a bad result. Would it be possible to provide some context on this?

- This is not an issue, but in Table 5, results with "Llama 3 (8B)" are reported. While it is great to have a comparison of
different models, Llama 3 (8B) is known to be comparably worse than e.g. GPT-4. The results reported with this model are not really very informative. For more significant results, a more powerful model of the Llama series, e.g. Llama 3 (70B), or Llama 3.1 (70B), that is claimed to be on par with close-models, should have been used.

- On line 52 "However, these studies have mostly focused on simplified synthetic datasets"
Could the authors please provide some references in this regard?

- On line 150:  "The agent selects the optimal tool by minimizing both time and error"
Could the authors please explain why (and how) is time and error minimized?

- It seems to me that the reference "BPC (2023)" does not belong to a serious
scientific work. I am sure that other references could be used here.

- The reference used for Llama 3 seems to be incorrect, and is incorrectly spelled
as "Lamma" instead of "Llama". The technical report where these models
were reported can be found at https://arxiv.org/pdf/2407.21783.

- The presented work seems to have potential ethical issues that may have
not been addressed or at least mentioned in the paper. For instance, since
the authors propose a potential method for predictive medicine, it is
important to note that a method with poor accuracy could lead to wrong
diagnosis and treatment. Besides this, is there any potential risk
in LLMs leading to biases in genomic analysis?

---

### Note · Authors · 2024-11-24

**Comment:**

We thank all the reviewers for their reviews and valuable feedback.

**Withdrawal Confirmation:**

I have read and agree with the venue's withdrawal policy on behalf of myself and my co-authors.